# A Simulation Study to Assess the Factors of Influence on Mean and Median Frequency of sEMG Signals during Muscle Fatigue

**DOI:** 10.3390/s22176360

**Published:** 2022-08-24

**Authors:** Giovanni Corvini, Silvia Conforto

**Affiliations:** Department of Industrial, Electronics and Mechanical Engineering (DIIEM), Roma Tre University, 00146 Rome, Italy

**Keywords:** power spectral density, spectral estimation techniques, Welch method, Burg method, autoregressive model

## Abstract

Mean and Median frequency are typically used for detecting and monitoring muscle fatigue. These parameters are extracted from power spectral density whose estimate can be obtained by several techniques, each one characterized by advantages and disadvantages. Previous works studied how the implementation settings can influence the performance of these techniques; nevertheless, the estimation results have never been fully evaluated when the power density spectrum is in a low-frequency zone, as happens to the surface electromyography (sEMG) spectrum during muscle fatigue. The latter is therefore the objective of this study that has compared the Welch and the autoregressive parametric approaches on synthetic sEMG signals simulating severe muscle fatigue. Moreover, the sensitivity of both the approaches to the observation duration and to the level of noise has been analyzed. Results showed that the mean frequency greatly depends on the noise level, and that for Signal to Noise Ratio (SNR) less than 10dB the errors make the estimate unacceptable. On the other hand, the error in calculating the median frequency is always in the range 2–10 Hz, so this parameter should be preferred in the tracking of muscle fatigue. Results show that the autoregressive model always outperforms the Welch technique, and that the 3rd order continuously produced accurate and precise estimates; consequently, the latter should be used when analyzing severe fatiguing contraction.

## 1. Introduction

In the last years, the use of surface electromyography (sEMG) exponentially increased in a variety of contexts and applications such as clinical assessment [1], sport performance evaluation [2], gesture recognition [3], classification [4], and prosthesis control [5,6]. In fact, this non-invasive technique provides useful information on the state of muscles [7]. For example, by variables in the time domain related to the signal amplitude, such as envelope or Root Mean Square (RMS), information on timing of muscle activation and on muscular force can be obtained [8,9], while by frequency parameters, information on muscle physiology and on muscular fatigue [10] can be derived. Among the various pieces of information, the one related to muscle fatigue is certainly of extreme interest. Since muscular fatigue has been associated with electrical signs, such as an increase of the amplitude of the sEMG signal and a compression of its spectrum toward the low-frequency area [11,12], attention has been devoted to the detection of parameters able to outline this behavior. Thus, to investigate the variation in the frequency content of the power spectrum, Mean Frequency (MNF) and Median Frequency (MDF) have been proposed [13] because they have been demonstrated to be related to alterations of firing rate and recruitment patterns of Motor Units (MUs) [14] that occur due to metabolic changes during fatigue. These important spectral features can be extracted from the Power Spectral Density (PSD) of the sEMG signals. Nevertheless, due to the finiteness of the real signals, the power spectrum cannot be computed, but only estimated; hence, several estimation techniques have been developed, each with its own advantages and disadvantages [15]. In general, it is possible to assess the quality of the estimates by studying the bias and the variance of the estimators [16]; however, the estimates also seem to be affected by the specific implementation settings of the estimators, such as the length and the shape of the signal segmentation window [17,18], the number of segments used for estimation [18], the frequency distribution of the spectrum [19] and the model order in parametric approaches [18,20]. Farina and Merletti [20], as well as Clancy et al. [10], compared the performance of different estimation methods on the basis of the epoch length used to process the sEMG signals. However, to our knowledge, comparisons have not been extended to muscle fatigue conditions in which the spectrum of the sEMG signal may take on shapes quite different from those typical of non-fatigue protocols [21].

In fact, the last methodological works that focused on the comparison of different methods in assessing the spectral parameters is the one of Farina and Merletti [20], in which useful recommendations for the spectral estimation with the autoregressive model were provided (i.e., the use of the 10th order of the Burg method). As a result, all subsequent studies exploit such recommendations without considering that changes in the frequency content of signals might affect the spectral estimates. For example, the study by Zhang et al. [22] investigated the PSD estimation of non-stationary signals with a time-varying autoregressive model, but still using a fixed order for the parametric approach. Moreover, a recent study [19] showed that the spectral estimates extracted from sEMG are affected by the frequency content of the signals.

The aim of this work, thus, is to test whether and to what extent the results of previous studies [19,20] can be considered valid in the case of fatiguing contractions associated with frequency distributions different from those found in the case of no fatigue. Two techniques for the Power Spectral Density estimation are considered: the Welch method and a parametric approach based on the AutoRegressive (AR) model.

The methods are applied on several synthetic sEMG time-series, each with its own Time duration (T), but all with the same compressed spectral shape. Different amounts of white Gaussian noise, indicated by the Signal to Noise Ratio (SNR) variable, are added to the signals to simulate as much as possible real acquisition conditions [23].

The performance of the estimators is assessed on the ability to determine the spectral parameters (i.e., MNF and MDF) and is quantified through the Mean Absolute Error (MAE) and its variance. This error is used as the criterion to determine the most robust estimation approach with respect to (i) the spectrum content and shape, (ii) the time duration of the signals, and (iii) the level of noise. For the parametric approach, the order of the model is also studied as a factor of influence of the results.

The paper is organized as follows: the estimation techniques for the Power Spectral Density estimation are described; the experimental design is presented; the statistical analysis is explained. Then, the performance of the two compared techniques is presented in terms of the error committed in the extraction of spectral parameters. Finally, the discussion section comments the findings, and in the conclusion section, some guidelines are provided. 

## 2. Materials and Methods

In this section, the model used for generating the signals is illustrated. The two estimation techniques, which have been used for the performance comparison, are fully described, as well as the spectral parameters that are calculated from the estimated power spectra. Then, the error that was used for the assessment of the performance is explained, and finally the ANOVA tests for the statistical analyses are presented.

### 2.1. Simulation Procedure

The model proposed by Stulen and De Luca [24] was used to generate a set of synthetic sEMG signals. This model takes as input a zero mean white process with unit variance that is filtered by a band-pass filter of which the square modulus of the transfer function is:(1)$$P_{xx}\left(f\right)=\frac{k^{2}f_{h}^{4}f^{2}}{\left(f^{2}+f_{l}^{2}\right){\left(f^{2}+f_{h}^{2}\right)}^{2}}$$

In this way, *P_xx_* represents the ideal PSD, *k*^2^ is a scaling factor, *f_l_* and *f_h_* are the low and high cut-off frequencies, respectively, and *f* is the frequency that ranges from zero to half of the sampling frequency (*f_s_*/2), because only the positive part of the spectrum is considered. The number of spectral lines (*L*) in the range *0*–*f_s_*/2 depends on the duration of the analyzed signal. The model parameters were set as follows: *k* = 1 and *f_s_* = 1024 Hz. The two cut-off frequencies, *f_l_* = 20 Hz and *f_h_* = 40 Hz have been selected such that the ideal MNF and MDF had a value of 39.84 Hz and 30.95 Hz, respectively. This specific pair of cut-off frequencies was chosen as low as possible to generate consistent myoelectric signals presenting a substantial compression of the power spectrum shape towards the low-frequency area, thus simulating a strong level of muscle fatigue, as highlighted in [25,26]. Eight different types of sEMG signals were generated considering eight different durations as in [20]: (a) T = 250 ms, (b) T = 500 ms, (c) T = 750 ms, (d) T = 1000 ms, (e) T = 1250 ms, (f) T = 1500 ms, (g) T = 1750ms, (h) T = 2000 ms. For each type, 1000 realizations were generated. A further random 1000 realizations of white Gaussian noise were added to the sEMG signals. Four typical SNR conditions were simulated, from 5 to 20dB, as in [23,27]. The resulting myoelectric signals have the following form:(2)$$x_{n}=\overset{N}{\underset{j=0}{\sum }}g_{n}h_{n-j}+q_{n}\;\;\;\;\;\;\;\;\;\;\;\;\;\;\;\;\;\;\;\;\;\;\;\;\;\;\;\;\;\;\;\;\;\;\;\;\;\;\;\;\;\;\;\;\;\;\;\;\;\;\;\;\;\;\;n=0,\;1,\;\ldots ,\;N$$
where *N* is the number of samples, *g_n_* is a realization of white Gaussian noise used as input of the shaping filter *h_n_*, and *q_n_* is a further realization of white Gaussian noise. The two processes of noise, *g_n_* and *q_n_* were assumed to be independent. The filter *h_n_* was obtained by taking the real part of the inverse Fourier Transform of the amplitude spectrum, that is the square root of *P_xx_*(*f*), and its phase was reconstructed as the imaginary part in the Hilbert transformation of the logarithm of the magnitude, as explained in [24].

### 2.2. Mean and Median Frequency

Mean and median frequency were computed from the power spectral densities. MNF is an average frequency, which is computed as:(3)$$\mathrm{MNF}=\frac{\sum_{l\;=\;1}^{L}P_{l}f_{l}}{\sum_{l\;=\;1}^{L}P_{l}}$$
where *f_l_* is the *l*-th frequency, *P_l_* is the *l*-th line of the power spectrum, and L represents the total number of spectral lines in the positive part of the spectrum. 

MDF, instead, is the frequency that splits the sEMG power spectrum into two regions exactly equivalent in power [13], and it is defined as:(4)$$\overset{l_{\mathrm{MDF}}}{\underset{l\;=\;1}{\sum }}P_{l}=\overset{L}{\underset{l_{\mathrm{MDF}}\;}{\sum }}P_{l}=\frac{1}{2}\overset{L}{\underset{l\;=\;1}{\sum }}P_{l}\;$$
where *P_l_*, *f_l_*, and *L* are the same as above. When the spectrum is symmetric with respect to its center line (e.g., Gaussian), MNF and MDF coincide, but typically, when dealing with myoelectrical signals, the distribution of the power in the frequency domain is left skewed and therefore the MDF is lower than the MNF.

### 2.3. Techniques to Estimate and Compute the Power Spectral Density

In the following, the Welch method, which is a non-parametric technique that estimates the power spectral density directly from the data, and the autoregressive model for PSD estimation are presented [28].

#### 2.3.1. Non-Parametric Estimation

The periodogram is one of the most known non-parametric estimation techniques but, unfortunately, this estimator is not consistent because the variance of its estimate does not tend to zero as the number of samples increases. Consequently, improved versions, which aimed to solve this issue, have been proposed, such as the Bartlett [29] and the Welch method [30]. The first one solves the inconsistency problem dividing the total length of the signal into S segments, computing the periodogram in each segment and then averaging the results to obtain the final PSD estimate; the second method works in a similar way, but it further improves the resulting PSD estimate because it allows overlapping windows. In this way, the improvement comes from the greater number of windows (thus decreasing the variance of the estimate), as well as the reduction of the loss of information at the extremities of the window due to the effect of the Fourier transform. The S segments are obtained by multiplying the signal to a window function (whose length is smaller than the total length of the signal), which is translated over the entire signal with a fixed overlap of samples. Hence, the resulting power spectral density can be estimated as: (5)$${\hat{P}}_{xx}\left(f\right)=\frac{1}{S}\overset{S}{\underset{s\;=\;1}{\sum }}I_{m}^{\left(s\right)}\left(f\right)\;\;\;\;\;\mathrm{where}\;\;f=0:\frac{f_{s}}{m}:\frac{f_{s}}{2}$$
where *S* is the total number (13) of segments, and *I_m_^(s)^* indicates the *s*-th periodogram that is estimated on m samples according to the following equation: (6)$$I_{m}^{\left(s\right)}\left(f\right)=\frac{1}{U}{\left|\overset{M}{\underset{m\;=\;0}{\sum }}w_{m}x_{m}e^{-j2\pi mf}\right|}^{2}\;\;\;\;\mathrm{where}\;\;U=\frac{1}{M}\overset{M}{\underset{m\;=\;0}{\sum }}w_{m}^{2}$$
with *M* being the total number of samples of the window, *w_m_* the window function, *x_m_* the signal, and *U* a gain factor. 

In this work, according to the results showed in Figure A1, the length of the window function was set to 25% of the total length of the signal, while the overlap was set to 25% of the length of the segment, so the total number of segments *S* was equal to 13. The zero-padding technique was applied to all the windows such that each periodogram was estimated on a total number of samples equal to the length of the entire signal. Results from a previous study have shown that the Tukey window function, also known as tapered cosine, outperformed other window functions in MNF and MDF assessment [19]; for this reason, this window was selected for the implementation of the Welch algorithm. 

#### 2.3.2. Parametric Estimation

For the parametric estimation, Autoregressive-Moving Average (ARMA) models are the most known. This parametric approach allows to estimate parameters of a mathematical model that can generate (and forecast) future samples by a linear combination of present and past inputs, and its past output. Autoregressive is a special case of the ARMA model and it is the most widely used for spectral estimation [20]. We used the Burg method [31], which estimates the model parameters directly from the measured data minimizing the prediction error that is generated by the difference of the actual output of the model and the real value of the signals analyzed. Given an order of the model *p*, the Burg technique estimates only the reflection coefficients *a_pp_* to predict future samples of a signal according to the following equation:(7)$$\hat{x}\left(n\right)=-\overset{p}{\underset{z\;=\;1}{\sum }}a_{pz}x\left(n-z\right)$$
where *p* is the order of the model, *a_pz_* are the prediction coefficients that can be computed using the iterative Levinson–Durbin algorithm, and *a_pp_* are the reflection coefficients (obtained when the index *z* is equal to the order *p*) that can be obtained by the minimization of the forward and backward errors of the estimation [15]. As a result, the Burg method aims to simultaneously minimize the sum of both the forward and the backward errors via the Least Mean Square Error (LSME) criterion. The power spectral density is thus computed as:(8)$$P_{xx}\left(f\right)=\frac{\sigma_{z}^{2}}{{\left|1+\sum_{z\;=\;1}^{p}a_{pz}e^{-j2\pi zf}\;\right|}^{2}}$$
where $\sigma_{z}^{2}$ is the total error and *a_pz_* are defined as before. In this work, six different orders, heuristically selected between the 3rd and the 30th order to compare their performance in the implementation of the Burg method, were analyzed. Different orders have been compared because the results of a previous work [19] demonstrated that the optimal order differs for the MNF and MDF computation, especially when a compression of the spectrum started to be visible in the frequency domain. 

### 2.4. Statistical Analysis

For the statistical analyses, the MAE is computed as following: (9)$$\mathrm{MAE}=\frac{1}{C}\overset{C}{\underset{c=1}{\sum }}y_{d}-y_{c}\;\;\;\;\;\;\;\;\;\;\;\;\;\;\;\;\;\;\;\;\;\;\;\;\;\;\;\;\;\;\;\;\;\;\;\;\;\;\;\;\;\;\;\;\;with\;c=0,\;1,\;\ldots ,\;1000$$
where *y_d_* is the ideal value of MNF (or MDF), *y_c_* is the MNF (or MDF) value computed from the estimation technique, and c is the total number of generated signals. Descriptive statistics (mean and standard deviation) were computed for both parameters. Interaction effects among factors were investigated by performing three-way ANOVA considering the following factors:
*method*, 6 levels (Welch, Burg 3rd, 4th, 7th, 10th, 15th and 30th order)*duration*, 8 levels (250 ms, 500 ms, 750 ms, 1000 ms, 1250 ms, 1500 ms, 1750 ms, 2000 ms)*SNR*, 4 levels (5 dB, 10 dB, 15 dB, 20 dB)


When the interaction effect among the three factors was significant, we set the values of *SNR* and we computed a two-way ANOVA for each level of the *SNR* factor; in turn, if the interaction effect between the two other factors (*duration* and *method*) was significant, we set the values of the *duration* factor, and then performed one-way ANOVA on *method* for each level of the *duration* factor. On the other hand, when the three- and two-way ANOVA were not significant, the main effect with one-way ANOVA on the *method* factor was directly studied. In each case, when the main effect of *method* was significant, the Tukey’s HSD post-hoc test was applied. Statistical analyses were conducted in MATLAB and the significance levels were set at: * *p* < 0.05, ** *p* < 0.01, *** *p* < 0.001.

## 3. Results

In Figure 1, the ideal PSD as well as those estimated with the Welch and Burg methods are shown. In this study, the difference between the ideal and the estimated shape was not assessed because we are interested in the values of the spectral parameters for fatigue detection. In Figure 1a,b, the PSDs estimated from signals with time duration equal to 250 ms and 2000 ms, respectively, when the level of noise is very high (SNR = 5 dB) are shown; in Figure 1c,d, instead, the PSD come from signals with SNR = 20dB and duration 250 and 2000 ms, respectively.

In Figure 1a, it can be noticed that, when dealing with brief signals (T = 250 ms), neither the spectrum obtained with Welch nor those computed with Burg succeed in the approximation of the ideal spectrum shape (in black). In fact, low orders of Burg produced a spectrum shape truncated around 0–5 Hz, and thus they were not able to approximate the ideal shape. In the same way, orders too high (30th) and Welch failed to generate a well-shaped spectrum shape because their spectra contained one small peak in correspondence of the high peak of the ideal spectrum, while the largest peak could be found around 50–60 Hz. It seems that the 10th order of Burg had the most similar shape, even if its peak (around 45 Hz) did not coincide with the ideal one around 20–25 Hz.

When time duration (T ≥ 1000 ms) of the signal increased, as shown by one example in Figure 1b, some high orders of Burg method (15th and 30th) approximated the spectrum shape well, having the central peak in the same frequency range of the ideal one. The spectrum estimated by the Welch method shifted the peak towards the low-frequency area, but it started to exhibit more oscillation. On the other hand, Figure 1c,d showed the power spectra estimated from brief (T = 250 ms) and long (T = 2000 ms) signals, respectively, with low level of noise (SNR = 20 dB). As can be seen, these two figures are similar to those corresponding to low level of SNR, indicating that the SNR did not substantially influence the estimation of the power spectrum shape. The main difference could be seen in the approximation of the spectrum shape obtained by the 3rd and 4th order of the AR model with signals of brief duration (T = 250 ms): when the SNR was equal to 20dB, the shape started to approximate the ideal one with a smoothed peak (Figure 1b) instead of having a sharp peak (Figure 1a).

Then, a three-way ANOVA was computed on both MNF and MDF, and the result of the test are summarized in Table 1. While no significant three-way interaction among *method*, *duration*, and *SNR* was visible in the study of the MNF, a statistically significant three-way interaction effect (*p* < 0.05) among these three factors could be seen when dealing with the MDF.

### 3.1. Mean Frequency

For the MNF, no significant three-way interaction effect among *method*, *duration*, and *SNR* was found, but there were significant two-way interaction effects between the following pairs of factors: *method* and *SNR*, and *duration* and *SNR*. The SNR influenced the estimate of the MNF producing substantial errors that were significantly different from one level to another independently from the estimation method, passing from an error of about 50 Hz when SNR was equal to 5 dB to an error of about 3 Hz when SNR was equal to 20 dB. The *duration*, instead, influenced the precision of the estimate: as the duration increased, the variance of the error decreased. However, since we are interested in finding the more robust method for the estimation, one-way ANOVA was performed on the *method* factor for each level of the *SNR* and for each level of the *duration* factor. Each of the test results was significant (*p* < 0.0001) and thus post-hoc tests were performed on the *method* factor. The results in Figure 2 show that the 3rd order of Burg outperformed Welch method and all the other orders of Burg (*p* < 0.01), except for one case: when SNR was equal to 20 dB, the mean of the MAE between the 3rd and all other orders were not statistically different (*p* = 0.99).

Results in Figure 3 are similar to those in Figure 2, but they represent the case when signals had T = 2000 ms. By comparing these results with those in Figure 2, it can be seen how the increase in the time duration of signals reduced the variance of the error, improving the precision of each method. Also in this case, the 3rd order of Burg outperformed Welch and all the other orders of Burg (*p* < 0.01) except when SNR was equal to 20dB: in this case, the difference between the 3rd and the 4th order was not significant (*p* = 0.99).

In general, Burg outperformed Welch especially when SNR was very low (SNR = 5 and 10 dB). As soon as the SNR increased, the difference between the two methods decreased, and even if significant, the difference between the best order of Burg and Welch was less than 1 Hz when SNR = 20 dB, for both, brief and long signals (T = 250 and T = 2000 ms, respectively). These results can be easily visualized in Figure 2 and Figure 3, where the MNF errors are reported for six orders of Burg and for Welch method. Each subplot in the figures corresponds to the analysis performed by setting one value of the *SNR*.

### 3.2. Median Frequency

For the MDF, three-way ANOVA revealed that there was a significant interaction effect (*p* < 0.001) among *method*, *duration*, and *SNR*, as shown in Table 1. Therefore, we set the value of the *SNR* factor, and a two-way ANOVA was performed, considering the interactions between the *method* and the *duration* for each level of *SNR*. The results, summarized in Table 2, suggest that there was always a significant interaction effect (*p* < 0.001) between the *method* and the *duration* factor.

Therefore, one-way ANOVA tests were performed on the *method* factor for each level of the *duration* factor, in turn computed for each level of *SNR*. All the one-way ANOVAs were statistically significant (*p* < 0.001), and thus post-hoc tests were executed to find out which level of the *method* factor produced the minimum error and had the best performance. In Figure 4, results obtained from signals with SNR = 5 dB are shown.

We can see that the lowest order of the Burg model outperformed the Welch method and all the other orders (*p* < 0.05) for every time duration of the signals; the same difference with Welch and all other orders was still present for the 4th order. For brief signals (T = 250 ms), the Welch method produced similar error to the 7th and 10th order (*p* > 0.05), while by increasing the time duration of the signals, it slightly reduced the error, producing comparable results (*p* > 0.05) to those obtained with higher orders (15th and 30th). Quantitatively, the mean difference between the 3rd and the 4th order was about 2–3 Hz, while between the 3rd order and higher ones and Welch was about 4–5 Hz. In general, the error produced by the best method (3rd order) was about 5 Hz when signals were very brief (T = 250 ms) and it decreased as well as the duration of signals increasing, reaching the initial error of about 2.5 Hz. This decrease (2.5 Hz) was found for each considered technique, indicating that longer signals allow to have a better frequency resolution.

In Figure 5, instead, it is possible to see the results obtained when SNR was equal to 20 dB. In general, the minimum error was produced by the 15th order of the AR model. When the signals had brief duration (T = 250 ms), the difference between the 15th order and the other levels of *method* factor were not significant, except when compared to the 30th order (*p* < 0.05). When the time duration of signals started to increase (T > 500 ms), the 15th order produced the minimum error, whose difference was statistically significant (*p* < 0.05) with respect to Welch and all other orders except for the 3rd and 4th order. Quantitatively, the mean difference between the 15th order and the other levels of the *method* factor was about 0.7–1 Hz for brief signals and it decreased to 0.2–0.5 for longer signals (T > 1000). In general, the error produced by the best method (15th order) was about 3 Hz when signals were very brief (T = 250 ms), but when the duration of signals increased, the total error reduced to 1.5 Hz. A decrease of about 1.5–3 Hz was found for each considered technique, confirming that better frequency resolution was obtained when working with longer signals.

## 4. Discussion

This study aimed to investigate the effects produced by the compression of the power spectral density, due to muscle fatigue, on the computation of the spectral parameters.

Although the results of this work will give some suggestion in the choice of the method to be used for the extraction of the spectral parameters, a few considerations need to be highlighted to correctly interpret the results. First, this study focused on a single power spectral shape representing an extreme case, that is severe muscle fatigue, which usually might be found in real data when analyzing muscle contractions until failure. Second, these findings, which have been extracted on synthetic sEMG, cannot be validated on real signals because the real value of the spectral parameters is unknown. As a result, the suggestions provided for the choice of the method could only ensure that the error in the spectral estimates will be limited depending on the analyzed condition.

Analyzing the spectra computed with the two estimation techniques, we noticed that the shapes of the spectra were not substantially influenced by the level of SNR (see comparison Figure 1a,c, or Figure 1b,d), except for the low orders (3rd and 4th) of the Burg method: this happens because a few parameters of the model are influenced by the high level of noise and are not able to approximate the shape of the spectrum to the ideal one being truncated at very low frequency. On the other hand, the time duration of the signal influenced the resulting shape: in fact, by increasing the length of the signal, we increased the frequency resolution. Both methods benefit from this increase in frequency resolution, but the Welch method still presented a lot of oscillations over the spectrum, which then affected the computation of the spectral parameter reducing the goodness of the estimates. Although there were no visible effects produced by the SNR on the shape, this factor highly influences the estimate of Mean and Median Frequency, as it can be seen in Table 1.

By analyzing the error produced in computing the Mean Frequency, we noticed that the time duration of the signal had no significant influence on the error, while the SNR had a great significant effect. When the level of noise was high (SNR = 5 and 10 dB), the error generated by the computation of the Mean Frequency was around 50 Hz and 19 Hz, respectively. Therefore, these huge errors are not acceptable, and we suggest avoiding the Mean Frequency use when dealing with noisy signals. Instead, if the level of SNR was high (SNR = 15 and 20 dB), the error around 5–7 Hz and 2–3 Hz, respectively, is still acceptable: results showed that the 3rd order of the Burg model is always the most performant in comparison with Welch and high orders of Burg. These specific results can only be considered valid when dealing with fatiguing contraction that are producing a harsh compression of the power spectrum. This finding is in contrast with the suggestion of always using a 10th order of the autoregressive model given by Farina and Merletti [20], but this is due to the fact that they considered a spectrum shape with the peak around 70–80 Hz that a 3rd order model is not able to approximate well. In contrast, if we need to analyze a compressed spectrum, the truncated shape obtained by the 3rd order (see Figure 1) produced a lower value of the Mean Frequency that was closer to the simulated ideal value. Therefore, in agreement with [19], we recommend decreasing the order of the autoregressive model to compute the Mean Frequency for tracking the development of muscle fatigue. However, the user should be very careful in using the Mean Frequency as an indicator of fatigue because it is highly influenced by the noise of signals, and the high level of error could lead to misleading results.

The analysis performed on the computation of the Median Frequency, instead, revealed that the obtained value was influenced by the interaction effects of the estimation technique, the time duration, and the amount of noise of the signals. As can be seen from Figure 3, even when the level of noise was very high (SNR = 5 dB), the error in the computation of Median Frequency was around 5–10 Hz, depending on the used estimation technique. As the time duration of the signal increased, the dispersion of the error around its mean, instead, was greatly mitigated. These findings confirm that the Median Frequency is more robust because it is less sensitive to noise than the mean frequency [24]. In fact, when the level of noise was low (SNR = 20dB), the error decreased to 4 Hz, and there were no significant differences between the errors produced by the different techniques. Moreover, as the time duration increased, with a consequent increase in the frequency resolution, there was a further reduction in the error down to 2 Hz. These results indicate that high accuracy in the computation of the Median Frequency can be obtained with both Welch and Burg techniques. On the other hand, the precision of the measure mainly depends on the time duration of the signals. For all these reasons, this study proposes to use a low order of the autoregressive model (3rd–4th) to estimate the Median Frequency when high level of muscle fatigue is to be assessed. Median Frequency should be preferred to the Mean Frequency if accurate measures are required even in presence of noise. In general, a normal sEMG shape of the spectrum could be estimated by high orders, (i.e., the 10th or the 6th, as stated in [20] and [19], respectively), but the order of the Burg methods need to be decrease to 3rd or 4th order as soon as muscle fatigue is approached.

## 5. Conclusions

This study aimed to investigate the effects produced by the compression of the power spectral density toward the low-frequency area; this variation in the frequency content is caused by the progression of muscle fatigue and influences the calculation of the Mean and Median Frequency. Two estimation techniques, Welch and Burg, were compared for the estimation of the power spectral density and the extraction of the spectral parameters. The purpose of this study, moreover, was to describe how the time duration and the level of noise of the signals affect the estimate of the power spectrum when it is harshly compressed in the low-frequency area.

The main finding of this work is that the Median Frequency should be preferred as indicator of muscle fatigue because it is less sensitive to noise than the Mean Frequency [24], always producing errors in the range of 2–10 Hz, according to the specific case. In fact, the use of Mean Frequency should be avoided when dealing with noisy signals (SNR <= 10 dB) because it produced enormous errors that are unacceptable.

In general, by increasing the time duration, and thus increasing the frequency resolution, improvements are produced in precision of the estimation, while increasing the SNR produces improvements in the accuracy of the estimates. Results suggested that the 3rd order of the autoregressive model produced accurate estimates analyzing fatiguing contractions, and therefore it is not necessary to use a high order (the 10th) as stated in [20], that will also increase the complexity and the time computation of the algorithm. These results, though, are valid when we are dealing with a power spectrum very compressed towards the low-frequency area due to the progression of high level of muscle fatigue; however, as stated in [19], the order of the autoregressive model for estimating the spectral parameter is not fixed, but it should be properly changed according to the frequency content of the spectrum that is examined, ranging from the 3rd order in presence of severe muscle fatigue to the 6th/8th order in normal conditions.

## Figures and Tables

**Figure 1 sensors-22-06360-f001:**
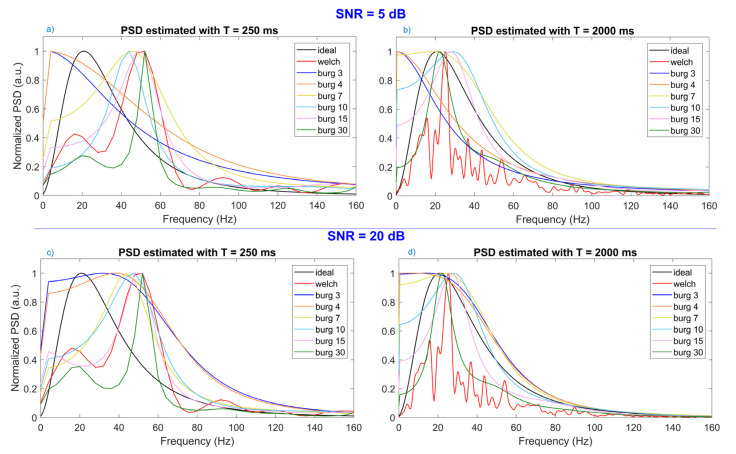
Representation of the ideal Power Spectral Density (PSD) together with power spectra estimated with Welch method and six different orders of the Burg method. (**a**,**b**) show the spectra estimated from signals whose Time duration (T) is equal to 250 and 2000 ms, respectively. These two figures were generated when Signal-to-Noise-Ratio (SNR) was low, that means there was a high level of noise. In the same way, in (**c**,**d**) power spectral densities, which were extracted from signals with time duration equal to 250 and 2000 ms, respectively, can be seen, but the SNR of signals was high, that is there was a low level of noise.

**Figure 2 sensors-22-06360-f002:**
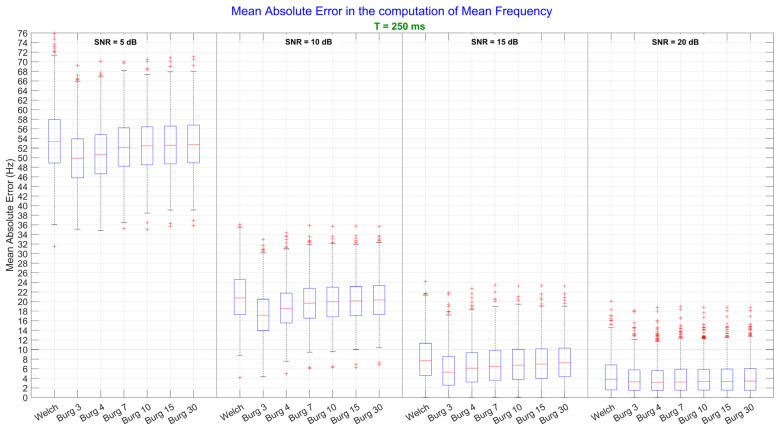
Mean frequency values computed from the power spectral density, which were estimated by Welch and Burg methods. The analysis performed on brief signals (T = 250 ms) is shown. In each subplot, a specific level of Signal-to-Noise Ratio (SNR) is represented. Since one-way ANOVA on the *method* factor was significant, post-hoc tests were performed: the significance level was set at *p* < 0.05.

**Figure 3 sensors-22-06360-f003:**
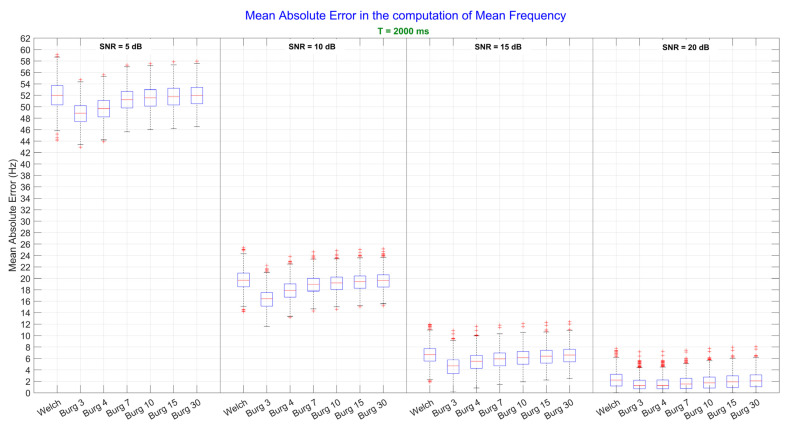
Mean frequency values computed from the power spectral density, which were estimated by Welch and Burg methods. The analysis performed on long signals (T = 2000 ms) is shown. In each subplot, a specific level of Signal-to-Noise Ratio (SNR) is represented. Since one-way ANOVA on the *method* factor was significant, post-hoc tests were performed: the significance level was set at *p* < 0.05.

**Figure 4 sensors-22-06360-f004:**
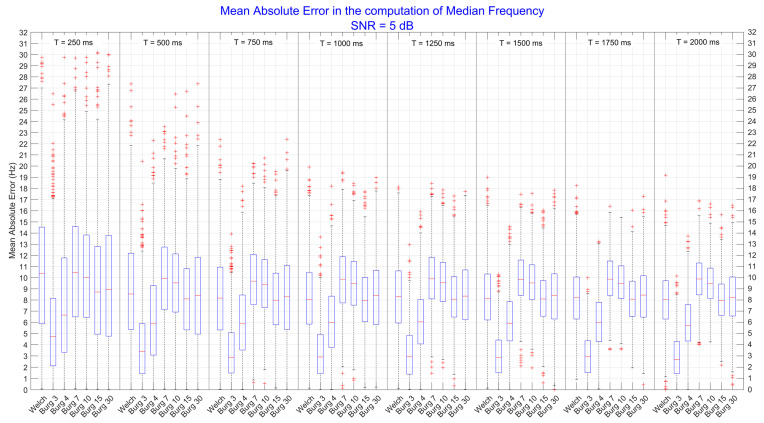
Mean absolute error computed on the median frequency parameters. All values were extracted from power spectral densities that have been estimated by Welch and Burg methods. The figure is divided into 8 subgroups (separated by vertical lines) representing the levels of the *duration* factor (e.g., T = 250 ms, …, and T = 2000 ms). Each represented subgroup was computed when the level of noise was high (SNR equal to 5 dB). The significance level was set at *p* < 0.05. The mean differences between the 3rd order and all other levels of *method* were statistically significant (*p* < 0.05).

**Figure 5 sensors-22-06360-f005:**
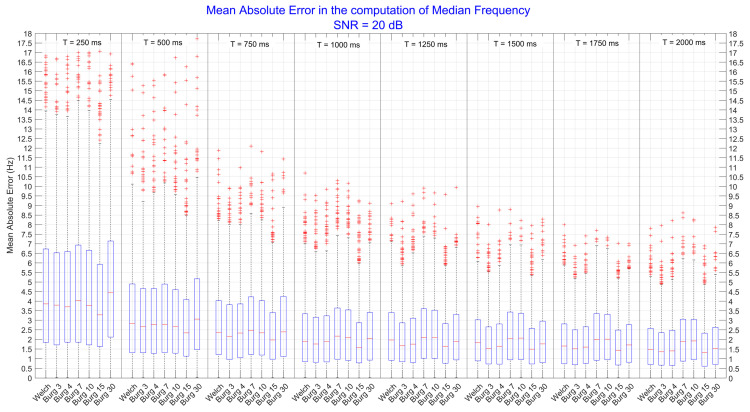
Mean absolute error computed on the median frequency parameters. All values were extracted from power spectral densities that were estimated by Welch and Burg methods. The figure is divided in 8 subgroups (separated by vertical lines) representing the levels of the *duration* factor. Each represented subgroup was computed when the signal-to-noise ratio was high (20 dB). The significance level was set at *p* < 0.05. The 15th order produces the minimum error but the mean differences with the error produced by the other orders and Welch were not significant (*p* < 0.05).

**Table 1 sensors-22-06360-t001:** Three-way ANOVA tests were performed to investigate the effects of the three factors on both Mean (MNF) and Median frequency (MDF). The three factors used were *method* (Welch and orders of Burg), *duration* (*T* varying from 250 to 2000 ms), and *SNR* (from 5 to 20 dB). The two-way and three-way interaction effects are shown starting from the 4th row of the table up to the 7th one, respectively. The significance levels were set at: * *p* < 0.05, ** *p* < 0.01, *** *p* < 0.001. Mean square, F-statistic, and *p*-value are provided. All the values less than 2 × 10^−16^ were indicated as 0, with the corresponding significance indicated by *** *p* < 0.001.

	Mean Frequency			Median Frequency		
Source	Mean Sq.	F	Prob > F	Mean Sq.	F	Prob > F
*method*	2.22 × 10^4^	2.95 × 10^3^	0 ***	1.52 × 10^4^	2.11 × 10^3^	0 ***
*duration*	5.67 × 10^3^	754.21	0 ***	1.79 × 10^4^	2.48 × 10^3^	0 ***
*SNR*	2.74 × 10^7^	3.65 × 10^6^	0 ***	3.39 × 10^5^	4.68 × 10^4^	0 ***
*method***duration*	10.05	1.33	0.07	89.06	12.30	0 ***
*method***SNR*	1.93 × 10^3^	257.06	0 ***	9.02 × 10^3^	1.24 × 10^3^	0 ***
*duration***SNR*	293.79	39.02	0 ***	277.71	38.36	0 ***
*method***duration***SNR*	1.74	0.23	1	21.57	2.98	0 ***
Error	7.52			7.23		

**Table 2 sensors-22-06360-t002:** Two-way ANOVA for the analysis of interaction effects between the *method* and the *duration* factors. Each ANOVA test was performed considering one level of *SNR* at time. The significance level is set at: * *p* < 0.05, ** *p* < 0.01 and *** *p* < 0.001. Mean square, F-statistic, and *p*-value are provided. All the values less than 2 × 10^−16^ are indicated as 0, with the corresponding significance indicated by *** *p* < 0.001.

	SNR = 5 dB		SNR = 10 dB		SNR = 15 dB		SNR = 20 dB	
Source	F	Prob > F	F	Prob > F	F	Prob > F	F	Prob > F
*method*	3.17 × 10^3^	0 ***	357.65	0 ***	71.34	0 ***	60.41	0 ***
*duration*	186.11	0 ***	629.21	0 ***	1.17 × 10^3^	0 ***	1.35 × 10^3^	0 ***
*method***duration*	5.48	0 ***	8.09	0 ***	3.75	0 ***	2.35	0 ***

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
