# Peer review of "A Simulation Study to Assess the Factors of Influence on Mean and Median Frequency of sEMG Signals during Muscle Fatigue"

_sensors, 2022, doi:10.3390/s22176360_

Round 1

Reviewer 1 Report

The authors present the article entitled “Effect of noise and epoch length on the computation of Mean and Median Frequency from synthetic sEMG signals simulating severe muscle fatigue”. However, It is not possible to extend my recommendation for publication according to the following concerns:

Include quantitative values in the abstract in order to highlight the findings.

Tables 1 and 2: Please check the guide for authors for table format.

Introduction section: The objective and novelty are not clear. Please, add at the end of this section, the objective of the article and the main contributions of the work. 

My biggest concern is that the novelty of the work is not clear because the bibliography presented in the state of the art (Introduction section) is too old. I suggest the authors check newer literature in order to find the novelty of the work.

How do the authors expect that this simulation could help and how does this improve in comparison with the conventional methods in real life?

Include a table that compares the findings of the work vs the actual works reported in the stat of the art.

line 28 can be justified with these fresher sEMG papers: A study of movement classification of the lower limb based on up to 4-EMG channels; A study of computing zero-crossing methods and an improved proposal for EMG signals; Support vector machine-based EMG signal classification techniques: a review

The article presents the following concerns:

  • The paper does not specify the objective or proposal in the abstract and the introduction. Be clear about that, for example, “this paper proposes, suggests, etc...” This section should be an objective representation of the article.
  • It's needed to describe the text structure in the final part of the introduction. Check the structure according to the guide for the authors of the journal.
  • I recommend giving an introduction between sections 2 and 2.1. 
  • The literature should be updated by using up-to-date references. More than 50% of references are from more than ten years ago.
  • Justify the relevance of the publication in the journal; it is necessary to include references from the journal.
  • Check the reference style according to the instruction for the authors of the journal.
  • Check “frecuency” in the text, because you capitalized “Frecuency” in some cases but not in others. Both styles are acceptable, but it’s best to use one style throughout your document. 
  • It apperas that “low frecuency” is missing a hyphen. consider adding the hyphen “low-frequency”.
  • Please add a nomenclature table to define variables and acronyms.
  • Apostrophes must be avoided
  • Paragraph three of "materials and methods" are almost the same as reference 9. Rewrite it.
  • The manuscript has a poor state-of-the-art. Try to include up-to-date references and good journals (avoid conferences)

The following misspelling should be checked:

  1. Line 16: “ muscle fatigue, have been used….” should be rewritten as  “ muscle fatigue, has been used….” (have-has Line 52 too).
  2. Line 21: “of the two methods is very similar working…” The word methods appears repeatedly in this text. consider using a synonym in its place. 
  3. Line 32: “and used as indicator because…” should be rewritten as “and used as an indicator because…”
  4. Line 49: “signal epoch length on…” should be rewritten as “signal epoch lengths on…” It seems that lenght may not agree in number with other words in this place. 
  5. Line 53: “by evaluating different level of…”  should be rewritten as “by evaluating the different level of”
  6. Line 102: “Periodogram is the most known…” should be rewritten as “The periodogram is the most known…”
  7. Line 112: “translated along the entire signal…” should be rewritten as “translated along with the entire signal…”
  8. Line 119: “previous study showed that Tukey window…” should be rewritten as “previous studies showed that the Tukey window…”
  9. Line 140: “computation in presence of muscle fatigue” should be rewritten as “computation in the presence of muscle fatigue.”
  10. Line 145: “the  estimation technique  and  c  is the  total  number  of  generated  signals…” should be rewritten as “the estimation technique, and  is the total number of generated signals…”

Line 199: “level of SNR analysed…” should be rewritten as “level of SNR analyzed…” (Same case in lines 207, 262)

Reviewer 2 Report

Effect of noise and epoch length on the computation of Mean 2 and Median Frequency from synthetic sEMG signals simulating 3 severe muscle fatigue

The manuscript presents the comparison of two estimation methods (Welch and Autoregressive) according to the noise and epoch length. The works make an interesting analysis but I have some recommendations. In the following lines, you can find it:

-        Abstract: Would it be possible to use the same procedure with not synthetic EMG signals?

-        2.1. Simulation procedure: some parameters are defined as part of the model. Why have you chosen those parameters?

-        Line 97: misspelling “techniques, , are described”

-        Line 128: misspelling. You say app instead of apk.

-        Line 159-60: it is said “it can be seen in figure 1-a that the shape of the ideal spectrum evaluated for 159 signals with short duration (T = 250ms) is almost well approximated by the higher order 160 of Burg method” but there is not any kind of parameter that describes this approximation. Have you assessed some kind of parameter to measure the “approximation”?

-        Figure 1: it is not enough one example to show your results. I recommend plotting at least one graph for each noise level to compare with figure 2 and figure 3.

-        Table 1: it is not clear. What does method mean? What does mean “epoch length” in this table? Is it the sum of all the epoch lengths? Explain what is the four rows in the bottom (SNR* epoch length, SNR* method, Epoch length* method, SNR* epoch length* method).

-        Figure 2: although the figure summarizes the results, some kind of table that shows the value in some points, could make possible a better understanding of the matter. Some points seem to be very close and it is not possible to read the figure properly.

-        Table 2: it is not clear. The same as table 1. An explanation of the rows’ titles could help the reader to understand the results.

-        Line 114 – 118: A length window set to 25% and an overlap of 50% makes the sample size very small. It could be the reason why figure 1 displays the problems you describe in the figure.

-        Line 211 – 213: AR and Welch method move with a similar difference in all the cases and it doesn’t fix with your statement. Maybe a table that summarizes those results can support your results and help to understand your comments.

-        Line 215 – 216: Results when noise is 20 dB behave differently than described. Please check all the results.

-        Line 210 – 230: This paragraph must be moved to the discussion section.

-        Discussion and Conclusion: I recommend splitting this section into two. On the one hand the discussion and on the second the conclusion.

Reviewer 3 Report

General:

1.      Overall: While a lot of interesting simulation work is described, I am concerned that the simulation model is too limited (only one spectral shape) and the model parameters studied may miss some possibly obvious ways in which to improve the estimators.  In particular, the “best” parametric estimator of the mean was the 6th-order model (the lowest order tested); hence the authors likely should have tested lower orders.  The best parametric estimator of the median was the 14th-order model (the highest order tested); hence the authors likely should have tested higher orders.  And, some established variations to the Welch method (zero-padding of segments, additional overlap) likely should have been considered, as your results suggest that improvement may have been available.  Until such additional analysis is considered, it is hard to accept that authors’ conclusions as to the preferred manner in which to compute these spectral features.

2.      You have not provided any literature support showing that the one spectral shape that you used (and its resulting mean and median frequencies) corresponds to a generic shape for EMG from a fatigued muscle.  And, it would likely have been better to repeat this analysis using a non-fatigued spectral shape.  Most studies that include localized muscle fatigue study both un-fatigued and fatigued contraction.  Likely, users need an analysis approach that is useful for both.

3.      All of the references seem to only draw from the EMG literature.  There is a wealth of information in signal processing textbooks and the general signal processing literature on the effect of epoch duration, overlap, window selections (etc.) for the Welch method and on model order for the parametric approach.  It would seem useful to draw appropriate information from this literature, both in the Introduction and the Discussion.  This general knowledge should inform your specific results.

4.      From Figure 1, it appears that all of the parametric models have difficulty modeling the low power level at frequencies near 0 Hz.  I think that the DC power of the parametric model equals your sigma-squared value in equation (8), since I think that the denominator goes to a value of 1 at DC.  But, it would seem that this sigma-squared value needs to be set to best match power across ALL frequencies.  Hence, this model form may be a poor fit for this problem.  So, rather than evaluating different orders of all-pole models, why not consider models that include at least one zero (for example, such that a zero can be placed at DC)?  My point is that Figure 1 is pretty good evidence that an all-pole model is NOT a good fit for a spectrum that has no power at DC.

5.      Figure 1a (top) also shows some issues with using your implementation of the Welch method for such short epochs (250 ms).  With this epoch duration, your segment length would be 62.5 ms (25% of the total length).  This segment length corresponds to 64 samples at 1024 Hz.  Hence, your spectral resolution is only 16 Hz.  Given that the mean and median were near 30–40 Hz, a spectral resolution of 16 Hz seems quite poor.  But, you have other options when using the Welch method.  For example, you can zero-pad each segment, allowing the Discrete Fourier Transform to essentially interpolate for you in the frequency domain.  And, other simple options exist (e.g., additional overlap; although more overlap tends to have diminishing returns).  In short, the poor resolution, which is quite obvious in Figure 1a, perhaps should be addressed as a factor that is limiting performance.  Certainly, Figure 2 suggests that performance improved with epoch duration. But, the number of segments was fixed at each epoch length, suggesting that spectral resolution (which improved as your segment length increased) could be the defining factor.  Thus, your data have some evidence to suggest that zero-padding of segments could be useful.

6.      For the Welch method, you set the segment length to 25% of the total epoch duration.  In doing so, your segment length was different for each epoch duration.  It is possible that this parameter should be tuned to the epoch duration.  Can this factor be considered?

7.      You studied mean absolute error as your only metric of error.  Most studies are not that sensitive to the ABSOLUTE error in estimating mean or median frequency.  Rather, they are sensitive to CHANGES in this error.  So, it is likely that such studies require a PRECISE measure, but not an ACCURATE measure.  Your mean absolute error criterion seems to measure accuracy and not precision.  Thus, I wonder if the variance of the difference (between actual and estimated mean/median frequency) might be a more useful value to study than the mean.

Specific:

8.      There does not seem to be a reason to capitalize the words “Mean” and “Median” frequency throughout the manuscript.  Consider using lower case (unless the word starts a sentence, etc.).  Similarly, the word “Autoregressive” is not generally capitalized.  Also, “Power Spectrum Density”.  And, many other words in the text are capitalized as part of defining an abbreviation; the typical style is to not do so.  Please check journal style for Sensors.

9.      Abstract: It would help to introduce quantitative descriptors and results into the Abstract.  For example, what autoregressive order is considered “low” or “high”?  What range of epoch lengths were studied?  Can you quantitatively provide information on error, likely in units of Hz?  And, the statistical significance of these changes should be noted in the Abstract.

10.  L 34, “… which are responsible for muscle fatigue.”: This statements seems to imply that the firing rate and recruitment CAUSE muscle fatigue.  I do not understand such to be the case.  In fact, the cause of fatigue is likely quite complex and more metabolic in nature, with EMG changes being a side effect of the metabolic changes.  In any case, perhaps review and more carefully phrase this statement.

11.  L 39: Consider changing “… must be estimated …” to “… can be estimated ….”  Several metrics exist to assess performance.

12.  L 43: Since citation [10] only has two authors, it would typically be referred to in-line as “Farina and Merletti [10]”.  Similarly on Line 44, since citation [2] has three authors, it would typically be referred to in-line as “Clancy et al. [2]”.  Also, although I am not fully familiar with these works, if only real data were analyzed, then the “true” mean and/or median value would not be known.  Assuming that your simulation work DOES know the true value, you would have an advantage in assessing bias error.  Perhaps a useful point to add, again, IF these two cited sources only used real data.

13.  L 61: Do not believe that “sagomates” is an English word.  I assume that you INPUT a white process, since equation (1) shows a spectral shape that is not white.

14.  L 67, L72 and please fix throughout: There should be a space between the value and the physical unit, e.g., “1024 Hz”, “250 ms”.

15.  L 68: Suggest referring readers to Fig. 1.  Also, I am not sure as to what are typical mean and median frequency values for a muscle that is severely fatigued (i.e., near failure).  Can you cite the literature and demonstrate that these resulting mean and median frequency values are realistic?  They seem quite low.

16.  L 77: Can you provide citations to support use of SNR values of 5–20 dB?

17.  Equation (2): It does not seem robust to use lower case “n” as both the sample index and a noise realization.

18.  L 81: If the noise sources are independent, I believe that they must also be uncorrelated.  Hence, the weaker assumption of uncorrelated is not needed.

19.  Equation (2): It is a bit unclear, but is seems filtering g_n by h_n–i produces the noise-free EMG signal, and n_n is the additive noise term.  A more explicit statement to this effect seems to be needed for most readers to follow your modeling.  Also, given the short duration of some of your epochs (as short as 250 ms), I would expect that the start-up transient of the FIR filter in equation (2) to be a problem.  Did you “pad” your durations, and then delete the startup transient?

20.  L 82, filter h_n: Did you determine h_n analytically using the inverse Fourier Transform?  If so, the model in equation (1) appears to be IIR, so how did you truncate the inverse transform so as to form the FIR filter coefficients?  If you determined h_n numerically, please provide methods, again describing how you formed a finite-duration impulse response.  Finally, how do you know that the realized h_n actually produces a random sequence with the specified spectrum shape, mean frequency and median frequency?  Especially if you used a numerical approach and truncated the impulse response, the achieved h_n will differ from the true h_n based on equation (2).

21.  L 84: I’m not much familiar with the Hilbert transform, but I am concerned that you used the imaginary part of a transform to reconstruct phase.  Do you need to explain more to the reader --- or perhaps add a citation to where the reader can find how to make this phase assignment?

22.  Equation (4): I don’t think that it is technically correct to use “MDF” as an index in the sums.  MDF is a frequency, in Hz, and need not be an integer.  The sums are indexed with integers, which have no physical units.

23.  L 106–121: This material lacks citations to the general Bartlett and Welch methods.  Some of this material seems a bit incorrect and some variables may be re-used.  For example, on L 106, you define “N” as the total number of samples and “S” as the number of segments.  On L 107, you state that each segment will have k = N/S samples.  This statement is only true if the segments are contiguous (which, I believe, is the Bartlett approach and not the Welch approach).  But, text on L 118 states that you used 50% overlap (Welch method). If you use overlap, the segment length is not N/S.  As another example, equation (5) seems to use “k” as the frequency index.  But, L 107 already defined “k” as the epoch length.  These usages seem inconsistent.  As one last example, I believe that the Welch method has a gain adjustment to account for power lost when applying the window function.  You do not show this adjustment.  (I believe that this gain adjustment would NOT change the value of the mean or median frequency, but it is part of the Welch method.)

24.  L 117: Does a window length of 25% of the total length and an overlap of 50% of the segment length (i.e., 12.5% of the total length) mean that you had 7 segments for each data set?

25.  L 128, 130: It is unclear what is meant by “a_pp” (as opposed to “a_pk”), since “a_pp” does not appear in equation (7) or (8).

26.  Equation (8): This equation also looks incorrect in some manner.  The left hand side shows the power spectrum to be a function of “f”, but “f” does not appear on the right-hand side.  Instead, the right-hand side seems to have frequency index “k”.  But, time index “n” also appears on the right (and is not the summing index, so remains after the sum over “l” occurs).  There should not be a time component to the spectral estimate.

27.  Figure 1: The titles seem to list the duration in seconds, whereas I believe they should be milliseconds.

28.  Figure 2: I know that a lot of plots are shown, but it would help to also get some idea as to the variance of these results.  If they are similar for the six methods, perhaps you could plot one?  Same issue for Figure 3.

29.  Figure 2: Each of the plots show that the minimum parametric model order of 4 had the lowest AVERAGE mean absolute error.  Thus, you have not shown that you have found the optimal order.  Lower orders should probably be tested.

30.  Figure 2 and corresponding statistics: Although you found the 6th-order parametric model to be best, ALL of the models have very poor performance when T = 250 ms.  Since the true mean is near 40 Hz, these errors (> 50 Hz) are nearly DOUBLE.  I would consider such estimates as unacceptable, but your text does not seem to reflect this concern.  Said another way, it does not seem to matter that the 6th-order parametric is best when T = 250 ms, since NONE of the estimates seems acceptable.

31.  Figure 3: As noted above, the Welch method had a spectral resolution of 16 Hz for the T = 250 ms epoch duration.  That resolution could certainly limit performance.  Also, the best parametric method consistently was the highest-order model.  So, again, it would probably help to see results at even higher orders, as results might continue to improve.

Reviewer 4 Report

Overview

The authors investigated the effect of noise and epoch length on the computation of Mean and Median Frequency from synthetic sEMG signals simulating  severe muscle fatigue. This study aimed to compare two estimation methods, Welch and Autoregressive, and to evaluate how epoch length affects their performance.

I have some suggestions for authors to improve the manuscript.

Specific comments

Abstract

-Line 14: The purpose of the study is not clearly understood. Replace with “This study aimed to compare two estimation methods, Welch and Autoregressive, and to evaluate how epoch length affects their performance”.

Keywords

Key Words must be different from the words in the title to optimize the search for the manuscript on the web. Replace “Mean Frequency, Median Frequency, epoch length” with other relevant Key Words.

Introduction

The introduction summarizes the existing literature in the field correctly. The gap in the literature is described. To highlight the purpose of the study, at line 48, I recommend that you start a new paragraph with the sentence "Therefore, in this work, results obtained...".

Materials and Methods

The methodology is clearly explained.

Measurements and procedures are correct.

The statistics are appropriate.

Results

Figures and tables must be independent of the text. The reader must understand them without having to read the rest of the manuscript.

Table 1 and 2:

-Please insert the full name of the acronyms used in the table in the notes.

-What does 0 *** mean? Either insert the three asterisks and indicate in the notes what it means (p <0.001) or write the p-value with precise values in the table (recommended).

Figure 1, 2 and 3:

Enter the meaning of all acronyms in the table in the caption. In the caption itself, acronyms can be avoided.

Discussion

In my opinion, the discussions need to be rewritten. It presents serious methodological errors.

There are no bibliographic references that confirm or refute what was found with this study.

In the first paragraph, describe the purpose of the study, the main results obtained and what is new.

Then describe how your study supports or criticizes current knowledge by comparing your findings with those of other scholars (citing them). Discuss how your findings support or challenge the paradigm.

At the end of the discussions describe what the study's limitations (not generic limitations) and strengths are. Explain why your results are robust.

Write the conclusions with an additional paragraph.

Conclusions

Restate your main findings and the novelty of the paper according to the current literature to help better readers understanding how this paper is different from other already published (Take-home message)

What are the consequences  of  the  study?  What  direction  should  research  work  take  in  this  area?  (Point  out  unanswered questions and future directions/implications). 

References

Bibliographic references are not enough. I advise authors to look for additional studies to cite. You should compare your findings with those of other scholars (to be cited) and describe how your study supports or criticizes their findings (discussion section).

Round 2

Reviewer 1 Report

The manuscript is ready for publication 

Author Response

Thank you for all the useful comments that allowed us to improve our manuscript.

Reviewer 2 Report

A simulation study to assess the factor of influence on Mean and Median Frequency of sEMG signals during muscle fatigue

The manuscript has improved since the previous version and the authors have taken into consideration the suggestions. It reflects a lot of effort to get it.

Abstract and introduction: It is needed to highlight the novelty of the work. In many paragraphs, it seems that you take the idea of other authors and copy it. Please add a main objective and some hypotheses at the end of the introduction to show your aim in this manuscript. Also, I suggest introducing novel works that give background to the manuscript.

Bibliography: there are many references too old. I recommend checking the literature and introducing some novelty manuscripts.

According to a question in my last review. You said this analysis can be done only with synthetic EMG signals because with real data you do not have the value of mean and median frequency to develop de comparison, but these results could make possible to use the most appropriate method with a real EMG signal? Are there limitations to use it in real life?

Author Response

The replies to the comments of the reviewer are attached in the Word file.

Reviewer 3 Report

General:

1.      Overall, the manuscript improves a great deal since the initial version.  A number of errors in description have been corrected and several methodical concerns have been addressed in detail.  Thank you for your efforts.

2.      You seem to have very complex statistical results due to the interactions.  In any case, your method of resolving the interactions do not seem to be appropriate, statistically.  I am NOT an expert in this area, but, consider the mean frequency results as an example.  You found NO three-way interaction between your factors method-duration-SNR.  But, you DID find two-way interactions between method-SNR and duration-SNR.  Your approach then was to (L 259) perform a one-way ANOVA on the method factor for each level of the SNR averaging over all the levels of the duration factor.  But, you KNOW that duration interacts with SNR.  So, I do NOT believe it statistically robust to average across all levels of this factor.  The interaction precludes this approach.  In short, you have two, two-way interactions, which end up involving all three factors.  So, you cannot perform ANOVA, as I understand it.  You likely CAN revert to making comparisons between all pairs of factors. But, because there are 6 methods, 8 durations and 4 SNRs, you will have 6 x 8 x 4 = 192 comparisons.  Thus, you will need to correct for multiple comparisons (Bonferroni-Holm may be best).  Perhaps there are other statistical methods of which I am not aware, when you have these types of interactions.  But, I am convinced that averaging across the duration factor is not appropriate, since the interaction tells you that the sense/direction of the result varies across duration.  In other words, an interaction implies that the result may be statistically “higher” at some durations while it is statistically “lower” at other durations.  So, averaging across the duration factors obscures this observation.

3.      I would recommend writing in the Discussion and Conclusion with less certainty about the generalizability of your results.  Certainly, you CAN summarize your findings.  But, you studied a single prototype frequency shape.  And, it is perhaps not overly surprising that since your prototype shape was parametric, you found that your parametric model performed best.  (Your Stulen-DeLuca model shape seems IIR, with a double zero at 0 Hz due to the “f**2” term in the numerator.  Your parametric Burg technique was also IIR, but had no zeros.)  My point is that your results may well show that your parametric Berj technique was a better predictor of a parametric Stulen-DeLuca model than a non-parametric spectral estimate.  But, it is unclear how well this result will translate to real data (which may not best fit the parametric model) or to other spectral shapes.  Thus, the work is likely not the final word on this topic.  But, hopefully it does add substantively to the literature.

4.      Some of your most important statistical results seem to appear in detail only as text within the caption of certain figures (e.g., Figures 2 and 3).  It would seem best to provide this information in the main text.

Specific:

5.      L 237: A test for an interaction between the THREE factors method-duration-SNR is usually referred to as a “three-way interaction.”  Tests for interactions between PAIRS of factors are usually referred to as “two-way interactions.”  So, as written, this text is confusion.  I believe you are trying to say that there was NOT a three-way interaction, but you did find two-way interactions involving all factors (method-SNR and duration-SNR.  So, please distinguish in the text between a three-way vs. a two-way interaction.

6.      L 253: No THREE-way interaction.

7.      L 254: But, you did find TWO-way interactions.

8.      L 260: I believe it incorrect to average over all of the “duration” levels, since you have a two-way interaction between duration-SNR.  So, I don’t think that these statistical results are valid.  With this many interactions, you may need to revert to paired testing with correction for multiple comparisons (e.g., Bonferroni correction, or Bonferroni-Holm correction).  Tukey tests might also be appropriate on the pairs, as I believe they can/are adjusted for multiple comparisons.

9.      Figure 2: The statistical results described in the figure caption likely belong in the text.

Author Response

The replies to the comments of the reviewer are in the word file attached.

Reviewer 4 Report

The authors responded adequately to the reviewer's suggestions. The overall quality of the manuscript has also been greatly improved. I accept in the present form.

Author Response

(The authors gave the same response as above.)

Round 3

Reviewer 3 Report

General:  Revisions to the statistics and to the interpretations/limitations have addressed all concerns from the earlier reviews.  No further general comments.

Specific:

L 283:  Suggest change “thos” to “those”.